# The NoHoW protocol: a multicentre 2×2 factorial randomised controlled trial investigating an evidence-based digital toolkit for weight loss maintenance in European adults

Sarah Ellen Scott [ORCID],[1] Cristiana Duarte,[1,2] Jorge Encantado,[3] Elizabeth H Evans,[4] Marja Harjumaa,[5] Berit Lilienthal Heitmann,[6,7] Graham W Horgan,[8] Sofus C Larsen,[6] Marta Moreira Marques,[3,9] Elina Mattila,[5] Marcela Matos,[2] Marie-Louise Mikkelsen,[6] António L Palmeira,[3] Beth Pearson,[1] Lauren Ramsey,[1] Kirby Sainsbury,[4] Inês Santos,[3] Falko Sniehotta,[4] Carol Stalker,[10] PJ Teixeira,[3] R. James Stubbs[1]

For numbered affiliations see end of article.

**Correspondence to**
Dr R. James Stubbs;
r.j.stubbs@leeds.ac.uk

## ABSTRACT

**Introduction** Obesity and associated diseases place a severe burden on healthcare systems. Behavioural interventions for weight loss (WL) are successful in the short term but often result in weight regain over time. Self-regulation of eating and activity behaviours may significantly enhance weight loss maintenance (WLM) and may be effectively augmented by contextual behavioural approaches to emotion regulation. The NoHoW trial tests the efficacy of a theoretically informed, evidence-based digital toolkit using a mobile-enabled website, activity trackers and Wi-Fi scales for WLM aiming to target (1) self-regulation and motivation, and (2) emotion regulation in adults who achieved clinically significant (≥5%) WL in the previous 12 months (initial body mass index (BMI) ≥25 kg/m$^2$).

**Methods and analysis** The study is an 18-month, 3-centre, 2×2 factorial single-blind, randomised controlled trial, which recruited 1627 participants achieving ≥5% WL between March 2017 and March 2018. Participants are randomly allocated to one of four arms: (1) self-monitoring only (self-weighing and activity tracker), (2) self-regulation and motivation, (3) emotion regulation or (4) combined self-regulation, motivation and emotion regulation. Participants attend four clinical investigation days at 0, 6, 12 and 18 months and are instructed to use the digital toolkit for 18 weeks during the first 6 months and at their discretion for the remaining 12 months. The primary outcome is change in weight (kg) at 12 months from baseline. Secondary outcomes are body composition (eg, bioimpedance analysis), health biomarkers (glycated haemoglobin, lipids, blood pressure, hair cortisol), dietary intake, physical activity, sleep, motivational, self-regulatory, emotion regulatory moderators/mediators of WLM, engagement, user experience, acceptability and cost-effectiveness of the interventions.

**Ethics and dissemination** Ethical approval was granted by Institutional Ethics Committees at the Universities of Leeds (17–0082; 27 February 2017), Lisbon (17/2016; 20 February 2017) and Capital Region of Denmark (H-

## Strengths and limitations of this study

► The first project to develop and evaluate (via randomised controlled trial) an information and communications technology toolkit combining continuous tracking of physical activity and body weight with theoretically informed, evidence-based digital interventions targeting self-regulation and motivation, and emotion regulation in a 2×2 design, to target long-term weight management.

► The 2×2 factorial design is a step forward from a conventional two-armed trial to determine the effect of self-regulation/motivation, emotion regulation interventions and their interactions on weight and health outcomes.

► Analyses of primary outcomes will test the hypotheses that self-regulation of energy balance behaviours improves longer term weight outcomes and that emotion regulation strategies help prevent weight relapse. Moderator analysis will determine if participant psychological/behavioural profiles predict response to self-regulation or emotion regulation interventions. Mediation analyses will examine mechanisms of action by which self-regulation and emotion regulation-based approaches affect energy balance behaviours, weight and health.

► Participants are blinded to study allocation, but they can access similar interventions from other real-world sources.

► There was no zero-intervention control. The active control arm includes self-monitoring of physical activity and weight but the inclusion of the Fitbit devices across all arms potentially contaminates or diminishes the effect of self-regulatory components. Social support was excluded as a potential contaminant of the digital intervention.

16030495, 8 March 2017). Results will be published in scientific journals.
**Trial registration number** ISRCTN88405328.

## INTRODUCTION

Overweight, obesity and associated diseases are key societal challenges. Obesity prevalence accounts for 3%–8% of health costs and 10%–13% of deaths in Europe.[1] This translates into €59 billion a year in direct, and an estimated €118–€236 billion in indirect, healthcare costs.[2] These costs are set to rise in parallel with obesity prevalence.[3] Approximately 40% of the adult population in Europe report engaging in at least one weight control attempt in the previous 12 months.[4] Existing community-based programme support initial weight loss (WL) but are subject to high attrition or weight regain, limiting longer term effectiveness.[5 6] The obesogenic environment and asymmetry of human energy balance (EB) regulation facilitates weight gain, while society criticises people suffering from overweight and obesity, leading to stress and negative emotions, which can undermine weight loss maintenance (WLM) attempts.[7 8] Effective support for WLM or the prevention of weight regain needs to address sustained changes in EB behaviours (dietary, physical activity and weight control) and additional emotional/psychosocial challenges that may undermine planned healthy behaviours.

### Self-regulation and motivation

Core features of effective WLM interventions include behaviour change techniques in line with self-regulation theories such as (1) *self-monitoring* of weight and behaviour; (2) *goal-setting*: agreement of clear weight targets/trigger points for weight control efforts; (3) feedback on behaviour and weight; (4) *action plans* for weight control through dietary and physical activity behaviours; and (5) *plans* to cope with *risk factors* for weight regain and relapse prevention (eg, problem solving).[9–15] Further, self-determined autonomous motivation has been consistently associated with change in EB behaviours for obesity management.[16 17] Interventions rooted in Self-Determination Theory may offer a promising approach to sustained EB and weight control.[12 17] A recent systematic review exploring psychological mediators of sustained effects in diet and lifestyle obesity interventions found that higher levels of autonomous motivation, self-efficacy, self-regulation skills, flexible eating restraint and positive body image were mediators of medium-term to long-term weight control.[18] The combination of motivational and volitional processes of EB behaviours can contribute to effective WLM; that is, it appears those who are adequately motivated and who implement intentions tend to both initiate and maintain such changes.[19–21] However, physical activity and dietary interventions based on current behaviour change theories characteristically achieve significant, but small and heterogeneous effects.[22] Additional psychological processes, such as emotion regulation, can potentially have incremental beneficial effects on the behavioural changes that promote WLM.[23]

### Emotion regulation

Individuals with obesity or those trying to lose weight can experience increased stress, depression, anxiety and weight-related stigma[24 25] which may undermine weight control attempts. Repeated attempts at WL followed by weight regain can have a negative emotional impact, leading to self-critical thoughts and negative emotions. For some, 'comfort eating' may be a means of coping with these negative experiences, potentially derailing strategies of planned behaviour.[26–28] Relapse prevention models include contextual behavioural (CB)-based emotion regulation strategies such as compassion, acceptance and mindfulness.[29–31] Evidence suggests that mindfulness and acceptance of undesired internal experiences (eg, negative emotions or self-criticism) and compassion for the self and others may help some individuals adopt adaptive behaviours that are congruent with their weight goals or values.[32–36] Mindfulness-based interventions show promise in changing obesity-related eating behaviours (eg, binge eating and emotional eating), lowering depression levels and improving WLM.[37] Augmenting current self-regulatory approaches with CB emotion regulatory approaches could lead to more effective WLM.[32–36]

### Digital behaviour change interventions

Recently, there has been interest in digital interventions to promote and maintain health.[38] Digital solutions, such as smartphone applications and wearables, can be effective in supporting WLM if they are evidence-based, apply a choice of behaviour change techniques with individualised feedback,[22 39 40] encourage the use of self-regulatory techniques (eg, self-monitoring) and develop intrinsic motivation.[41] Digital solutions are potentially cost-effective and scalable to large populations, which could engage citizens in healthcare innovations that are convenient and effective for weight management in the face of limited public budgets.[42] To date, most commonly available weight management applications miss key evidence-based approaches for longer term behaviour change.[43 44]

### Study objectives and hypotheses

The effectiveness of long-term evidence-based digital interventions and strategies to support WLM is limited.[45] The study's primary objectives are to evaluate whether using a new digital toolkit is effective for WLM by improving (1) self-regulation and motivation, (2) emotion regulation or (3) these factors in combination, compared with (4) control (generic toolkit content, regular self-weighing and Fitbit use), in 1627 participants at three European centres. We hypothesise that the combined intervention will be more effective for WLM compared with the self-regulatory or emotion regulatory interventions alone, which will also be more effective than the control group.

Secondary objectives are to determine (1) how the intervention affects health markers (eg, levels of glycated haemoglobin (HbA1c)) and body composition; (2) examine the intervention impacts on physical activity, sleep, dietary intake, depression, anxiety, stress, quality of life and well-being; (3) examine mediators of WLM, such as self-regulation (eg, planning capacity), motivation (eg, autonomous motivation) and emotion regulation

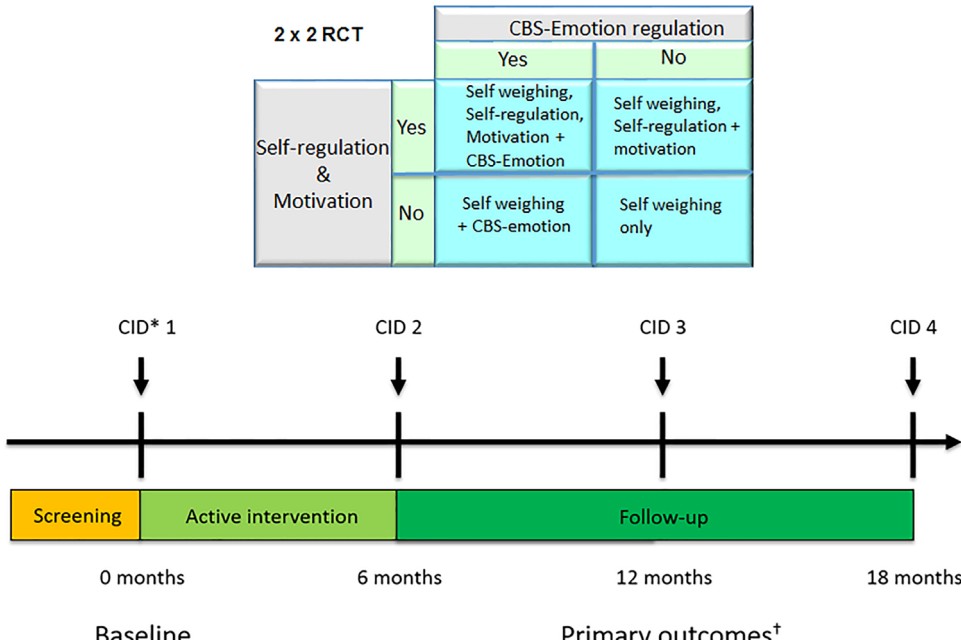

**Figure 1** Trial design. RCT, randomised controlled trial.

processes (eg, self-compassion); (4) analyse baseline moderators (eg, gender, body mass index (BMI)) to identify participants who are more responsive to motivational and behavioural self-regulatory or CB-emotion regulatory approaches for WLM; (5) conduct quantitative and qualitative assessment of user-experience, acceptability, engagement and dropout; (6) examine intervention cost-effectiveness; and (7) to determine the efficacy of individualised feedback from data on physical activity self-tracking.

## METHODS
### Trial design and setting
The NoHoW trial is a three-centre (University of Leeds (UK), The Parker Institute (Denmark) and University of Lisbon (Portugal)) 2×2 factorial, randomised, single-blind, controlled trial testing the proof-of-concept of a digital toolkit for WLM. The study duration is 18 months with a follow-up at 6, 12 and 18 months postbaseline (figure 1). The study is conducted between March 2017 and September 2019 in academic research institutes. In total, 1627 participants (~542 per centre) were enrolled and randomised into one of four intervention arms: (1) active control arm (consisting of generic toolkit content, self-weighing and activity trackers, only); (2) self-regulation and motivation arm; (3) emotion regulation arm; and (4) combined motivation/self-regulation and emotion regulation arm (figure 1). The protocol is harmonised across trial centres using Good Clinical Practice guidance, research-grade translation/backtranslation

of trial materials, two training workshops and weekly trial management meetings. The trial registration data set is displayed in table 1. The final protocol (V2.1 20/09/2017) was approved by the Trial Steering Committee (TSC) and adheres to the Standard Protocol Items: Recommendations for Interventional Trials guidelines.[46]

### Participants
Table 1 describes the inclusion/exclusion criteria for the trial. Prior to randomisation, potential participants are asked to provide documented verification (by a health professional, WL counsellor/friend, WL programme record booklet, diary or smartphone app or before/after photographs[47]) that they have achieved a clinically significant WL of >5% during the previous 12 months.

### Study recruitment
The study recruitment and intervention schedule is outlined in table 2. Centre-specific recruitment strategies were adopted for 12 months (March 2017–March 2018) and included commercial WL programme (UK, Slimming World); the Copenhagen Municipality weight management services, Dieticians from the Danish Association for Dieticians and commercial slimming companies (eg, Sense, Henrik Duer and Per Nielsen); registered clinical dieticians/nutritionists who provide weight management services in Lisbon; leisure centres; and local/national media coverage and advertisements. All participants were directed to country-specific recruitment websites (http://uk.nohow.eu; http://dk.nohow.eu; http://nohow.fmh.ulisboa.pt) and completed an

**Table 1** WHO trial registration data set

| Data category | Information |
|---|---|
| Primary registry and trial identifying number | ISRCTN: 88 405 328 |
| Date of registration in primary registry | 16/12/2016 |
| Secondary identifying numbers | H2020-EU.3.1 Grant agreement ID: 643 309 |
| Source(s) of monetary or material support | European Commission, Horizon 2020—EU Framework Programme for Research and Innovation |
| Primary sponsor | University of Leeds, Leeds, UK |
| Secondary sponsor(s) | The Parker Institute, Copenhagen, Denmark<br>University of Lisbon, Lisbon, Portugal |
| Contact for public queries | Prof. James Stubbs (r.j.stubbs@leeds.ac.uk)<br>School of Psychology<br>Faculty of Medicine and Health<br>University of Leeds<br>Leeds<br>LS2 9JT<br>UK<br>+44 (0)113 343 3476 |
| Contact for scientific queries | Professor James Stubbs (r.j.stubbs@leeds.ac.uk) |
| Public title | NoHoW: Evidence-based ICT tools for weight loss maintenance |
| Scientific title | A 2×2 randomised controlled trial to evaluate the effectiveness of evidence-based ICT behaviour change tools for weight loss maintenance in overweight/obese adults after clinically significant weight loss. |
| Countries of recruitment | 1. University of Leeds (UK).<br>2. The Parker Institute (Denmark).<br>3. University of Lisbon (Portugal). |
| Health condition(s) or problem(s) studied | Adults aged 18 or older who have intentionally lost at least 5% of their body weight in the last 12 months, and initially had a BMI of 25 or more. |
| Intervention(s) | Participants are assigned to one of four interventions:<br>1. Self-monitoring only (self-weighing and activity tracker).<br>2. Self-regulation and motivation.<br>3. Contextual behavioural science-based emotion regulation (CBS-emotion regulation).<br>4. Combined self-regulation, motivation and CBS-emotion regulation. |
| Key inclusion and exclusion criteria | Inclusion criteria:<br>1. Aged 18 years or older.<br>2. BMI (prior to weight loss) of ≥25 kg/m$^2$.<br>3. Verification of ≥5% of weight loss in the last 12 months and remain 5% below their highest weight.<br>4. Ability to use a smartphone and have access to smart phone, tablet or computer with internet access and Wi-Fi.<br>5. Ability to use standing scales for weight measurements and must not be over 150 kg (the scale limit 158 kg)<br>Exclusion criteria:<br>1. Inability to give informed consent.<br>2. Lost weight due to illness or surgical procedures, including bariatric procedures.<br>3. Pregnant or breastfeeding.<br>4. Involvement in another research intervention study that confound with the aims of NoHoW (excluding local health interventions and weight management services).<br>5. Inability to follow written material or telephone conversations in the English, Danish or Portuguese language (depending on the centre).<br>6. Diagnosed with an eating disorder (eg, anorexia nervosa, bulimia nervosa or purging disorder).<br>7. Diagnosed with any condition that may interfere with increasing mild to moderate physical activities and which is unstable (ie, untreated or unable to be controlled by medication).<br>8. Recent diagnosis with type 1 diabetes.<br>9. Extensive planned travel, that is, moving away or long-term travel abroad (eg, more than 4 weeks).<br>10. Living in the same household as existing participant in the trial. |
| Study type | Interventional, 2×2 factorial randomised controlled trial. |
| Date of first enrolment | 20 March 2017 |
| Target sample size | 1600 |
| Recruitment status | Completed (31/03/2018), no longer recruiting. |

Continued

**Table 1** Continued

| Data category | Information |
|---|---|
| Primary outcome(s) | Weight (kg) |
| Key secondary outcomes | 1. Body composition: (eg, fat mass and fat-free mass) and waist/hip ratio. |
| | 2. Biomarkers: blood pressure, heart rate, cortisol (hair samples), HbA1c, cholesterol. |
| | 3. Behavioural: physical activity, sleep and dietary intake. |
| | 4. Psychological moderators and mediators of change for: self-regulation and motivation (eg, action planning and coping, self-efficacy); CBS-emotion regulation (eg, weight focused self-criticism, five dimensions of mindfulness); stress management (eg, perceived stress scale, depression, anxiety and stress); and quality of life and well-being. |
| | 5. Process evaluation |
| | 6. Cost-effectiveness analysis |

BMI, body mass index; ICT, information and communications technology.

online eligibility screener using Qualtrics. The eligibility screener categorised respondents as eligible, potentially eligible (wait-list) and ineligible. Eligible individuals were contacted for a telephone screening interview, provided with study information, medical history questions and the Physical Activity Readiness Questionnaire.[48] Eligible participants were invited to a clinical investigation day (CID) where informed consent was obtained by research staff before randomisation (online supplementary file 1). Participants are reconsented before each subsequent visit and excluded if they become ineligible.

### Public involvement

Members of the public that matched the study's inclusion criteria at the three trial centres were consulted via a survey to inform the recruitment strategies, trial conduct, questionnaires, assessment of burden and reviewing or testing the intervention content at all trial centres. During the toolkit development stage, the public were involved in two ways. First, a panel of 10 people, matching the trial criteria, tested the toolkit V.1 written in English

for 1 month in the UK. A mixed methods approach was used for the analysis of the data assessing technical maturity of the intervention and user experiences, through questionnaires (quantitative data), interviews (qualitative data), TK usage log data and data from wireless scales and activity trackers (Fitbit). These analyses will be published in a separate paper. The results were used to inform toolkit V.2. Toolkit V.2 was then tested for 18 months using a panel of 30 people (10 in each centre) starting 3 months before the launch of the trial. Trial staff regularly collaborated with the panel to identify technological glitches that could be corrected early on so as not to impact on the main trial participants.

### Randomisation and blinding

Participants were randomly allocated to treatment arms by researchers using an online trial administration portal. Adaptive stratified sampling using minimisation[49] was embedded in the portal using the R-programme (R Foundation for Statistical Computing, Vienna, Austria) by the study statistician GH. This minimises differences

**Table 2** Study recruitment and intervention schedule (Standard Protocol Items: Recommendations for Interventional Trials 2013 guidelines)

| | Study duration | | | | |
|---|---|---|---|---|---|
| | Recruitment | Baseline | 6 months | 12 months | 18 months |
| TIME POINT | t0 | t1 | t2 | t3 | t4 |
| RECRUITMENT | | | | | |
| Screening | X | | | | |
| Informed consent | | X | | | |
| Randomisation and allocation | | X | | | |
| INTERVENTION | | | | | |
| Fitbit device training | | X | | | |
| Intake24 (food diary) training | | X | | | |
| Toolkit training (after randomisation) | | X | | | |
| Arms 2–4 active intervention | | ◆ – – – – – – ◆ | | | |
| Arms 2–4 follow-up | | | ◆ – – – – – – – – – – – – – – – – – ◆ | | |
| Arm 1 (control) intervention | | ◆ – – – – – – – – – – – – – – – – – – – – – – – – – ◆ | | | |

in age, WL in the previous 12 months and current BMI between treatment arms and stratified by gender and centre. Participants were blinded to study allocation. The research team were not blinded due to the need to train participants in arm-specific toolkit versions. The statistician (GH) is independent from data collection but not blinded during analyses due to inclusion of the arm allocation number. Independent researchers analysing hair

cortisol are blinded to group allocation. No preliminary or ancillary analysis of outcome data between intervention arms were conducted prior to data-lock (12 months).

### Interventions

The toolkit is a mobile-enabled website, delivered as a portfolio of embedded tracking technologies, mini-apps and web resources (figure 2) accessible via computer, tablet

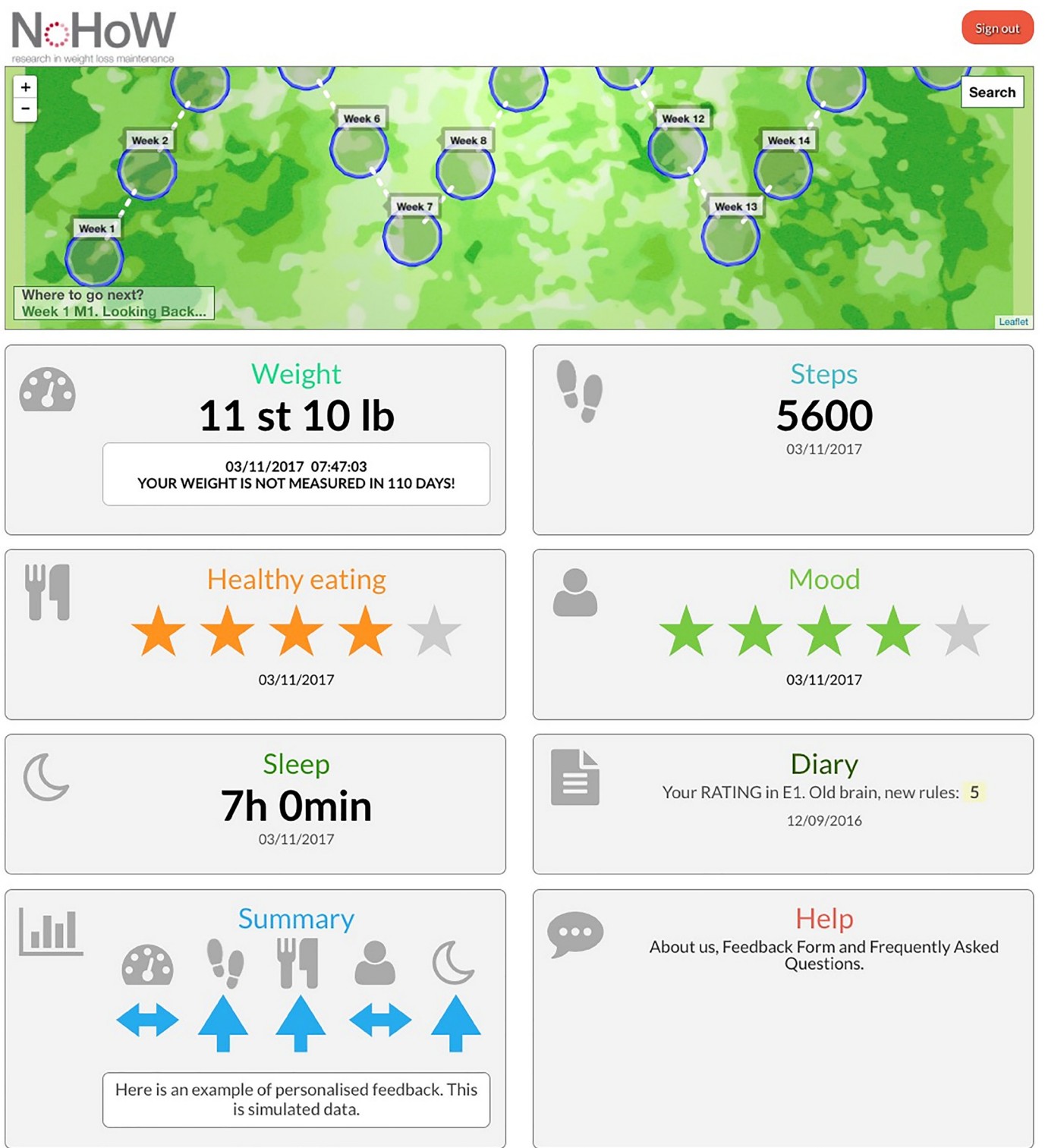

**Figure 2** Toolkit example (arm 4).

and mobile phone. The tracking technologies consist of an activity and sleep tracker (Fitbit Charge 2) and smart wireless body weight scale (Fitbit Aria). Participants were asked to wear the Fitbit Charge 2 at all times and to weigh themselves ≥twice weekly. All arms include instructions for self-weighing and use of activity trackers. Arms 2 and 4 include individualised weekly feedback about weight trajectories. Guidance for complex interventions was followed during the toolkit development specifying the intervention logic models and theory-driven behaviour change techniques,[50] which will be detailed in a separate publication.

### Arm 1 (active control, self-monitoring, only)

The active control arm includes a toolkit dashboard containing weight, steps and sleep graphs using data from the Fitbit devices, healthy eating and mood data (self-scored on a five-point scale), a diary to populate with notes from these five tiles and a help tile. The content is generic and does not relate to any theoretically informed, evidence-based behaviour change approaches.

### Arms 2–4

After randomisation, participants received standardised training from researchers during the first CID, specific to their arm allocation. Trial arms 2–4 include the same components as arm 1 but participants also receive a personal virtual road map (the 'NoHoW Journey') with intermediate destinations represented in that map by circles, each comprised of evidence-based theoretically informed thematic modules targeting self-regulation and motivation (arm 2), emotion regulation (arm 3) and combined self-regulation and emotion regulation (arm 4). Arm 4 includes the same number of thematic modules as arms 2 and 3. Each weekly module opens into submodules. The active intervention involves 18 weekly sessions delivered throughout a 6-month period. Users are encouraged to complete ≥1 submodule per week following the sequence in the road map, consisting of videos/audios, interactive graphs, text, images, questionnaires and mini-apps. Weekly arm-specific emails are sent during the active 6-month intervention phase, which provide WLM information, suggestions and reminders to visit specific content. The toolkit automatically sends an email notifying participants to slow down if they complete three sessions ahead of schedule. After 6 months, they are encouraged to revisit the toolkit for a further 12 months at their discretion.

Individualised feedback is a component of the self-regulation and motivation arms (arms 2 and 4) automatically provided by the toolkit in response to the individual's weight trajectory. Feedback is generated from the data extracted from three sources: (1) weight from the Fitbit Aria scales; (2) activity and sleep from the Fitbit Charge 2; and (3) user engagement with the toolkit, including day of the week. After daily usage patterns are analysed, feedback is displayed in the toolkit consisting of one of

a set of short statements (eg, "your weight management appears better when you are more active").

### Adherence

The toolkit sends automated weekly emails to participants to encourage engagement, prompt log-in and recommends sessions to complete (arms 2–4 only). The control group receives the same number of emails but with links to generic content (eg, topical diet/health information). Intervention adherence is continuously monitored by collecting information on the number of days since users last logged into the toolkit and the number of days since the last synchronisation between the Fitbit devices and data-hub. If no activity is detected after 21, 28 and 35 days, three attempts are made to contact participants by email. After 42 days of no activity, researchers telephone participants to resolve issues. Reasons for no activity are recorded.

### Assessments

The primary outcome is change in weight (kg) between baseline and 12 months (primary endpoint). Follow-up measures for WLM are completed at month 18 for comparability with other longitudinal studies.[22] All physical measurements are taken after a 10–12 hour overnight fast by trained research staff. Measures are taken at all four time points (baseline, 6, 12 and 18 months) except HbA1c, lipids and hair cortisol (baseline and 12 months only) and process evaluation measures (1, 3, 6 and 12 months only). The study schedule and information for all measures is in online supplementary file 2.

### Participant retention

Randomised participants are contacted at least 2 weeks before each visit and sent an email/text reminder. If participants deviate from planned assessments, three attempts are made to contact them by phone, email and a final written letter across 1 month. At least weight (primary outcome) is obtained if possible.

### Primary outcome
#### Change in weight (kg) at 12 months from baseline

Body weight (±0.1 kg) is measured using a Seca 704s instrument (SECA, Germany) in participants wearing light clothing. Participants are asked to weigh themselves at least twice weekly, in the morning after voiding and before eating using the Fitbit Aria scales for 18 months.

### Secondary outcome measures
#### Body composition and height

Multifrequency whole bioimpedance spectroscopy is measured by ImpediMed (SFB7, Queensland, Australia), which measures impedance over a spectrum of frequencies for the estimation of body composition. Hanai mixture theory equations and standard resistivity constants[51] are used to calculate fluid volumes to estimate total body water. Height (±0.1 cm) is measured with participants barefoot, using a Seca 704s instrument (SECA, Germany). See online supplementary file 2.

### Waist–hip ratio

Waist and hip measurements are taken according to the WHO's guidance. A tape measure is used to record the hip and waist circumference to the nearest centimetre.[52] The waist–hip ratio is calculated by dividing hip and waist circumference.

### Biomarkers
#### HbA1c and cholesterol

Fasted capillary blood samples are collected to determine HbA1c (mmol/mol, %), estimated average glucose (eAG, mol/L) and full lipid profiles, including total cholesterol, low-density lipoprotein, high-density lipoprotein (HDL), triglycerides, non-HDL and cholesterol/HDL (mmol/L) assayed using a benchtop analyser (Alere Afinion AS100 Analyser, Alere, Stockport, UK).[53]

#### Blood pressure heart rate

Systolic and diastolic blood pressure (BP) and resting heart rate (HR) (Microlife BP A2 Basic, Gentle Technology, Microlife, Clearwater, FL, USA,) are measured three times with the participant at rest in the sitting position. The mean value is recorded.

#### Hair cortisol

Hair samples are collected by cutting a small tuft of hair close to the scalp to measure the stress hormone, cortisol, and analysed following a previously described protocol[54] (online supplementary file 2).

#### Physical activity

Physical activity is measured by the self-reported International Physical Activity Questionnaire (IPAQ)[55] and Activity Choice Index (ACI).[56] Minute-by-minute physical activity data and heart rate are measured by the Fitbit Charge 2 for the study duration.

#### Sleep quality and quantity

The Fitbit Charge 2 estimates sleep quantity (hours/minutes) and quality (stages of sleep). Data are collected throughout the trial.

#### Dietary intake and eating behaviour

Four consecutive 24-hour-web-based dietary recalls, including at least 1 weekend day, are collected within 7 days of each visit using INTAKE 24.[57] Psychometric measures of eating behaviour are assessed at each visit using the Three Factor Eating Questionnaire-51,[58] the Problematic Eating Patterns Scale (unpublished scale), Eating in the Absence of Hunger Scale,[59] Intuitive Eating Scale-2[60] and the Binge Eating Scale.[61]

#### Well-being and quality of life

Quality of life and well-being are assessed using the 5-level EQ5D (EQ5D-5L)[62] and the Warwick-Edinburgh Well-Being Scale,[63] respectively.

#### Self-regulation

Self-regulation is measured using the Action Planning and Coping Planning Scales[64] and Action Control Scale.[65]

### Motivation

Satisfaction of psychological needs are assessed using the Basic Psychological Needs and Frustrations Scale.[66] Quality of goal contents is assessed using the Goal Content for Weight Loss Maintenance Scale.[67 68] Autonomous and controlled motivation are assessed using the Behavioural Regulation in Exercise Questionnaire (BREQ-3),[69] Regulation of Eating Behaviour Scale (REBS)[70] and Regulations for Weight Management Scale (adapted from the REBS).[70]

### Self-efficacy

Self-efficacy is assessed using the Self-Efficacy for Weight Maintenance Scale.[71]

### Emotion regulation and stress management

CB-emotion regulation and stress management constructs are measured using the Weight Focused Self-Criticism/Self-Reassurance Scale,[72] Weight Focused External Shame Scale,[73 74] the Compassion Engagement and Actions Scales,[75] Body Image Acceptance and Action Questionnaire,[76] Engaged Living Scale,[77] Mindful Attention Awareness Scale,[78] adapted Difficulties in Emotion Regulation Scale[79] and Experiences Scales (decentering subscale only).[80] Perceived stress is measured by the Perceived Stress Scale.[81] Anxiety and depression are measured using the Depression & Anxiety Stress Scales.[82]

### Demographics, weight history and participant traits

A profiling questionnaire, adapted from the DiOGenes study,[83] is administered at baseline only to measure trait-specific moderators of WLM including the following demographics: age, gender, country, education, marital status, employment and income. Also, smoking and alcohol use and number of times giving birth are recorded. Weight history includes height, current and highest weight in the last 12 months, successful and unsuccessful WL attempts and strategies, and psychological impact of living in a food abundant environment (the Power of Food Scale).[84] Self-determination traits are assessed using the Index of Autonomy Functioning[85] and Aspirations Index,[86] and self-regulation of appetite, hunger and satiety cues by the Intuitive Eating Scale-1.[87] Relapse prevention capability is measured using an adapted Relapse Prevention Scale.[88]

### Process evaluation and user experience

Questionnaires adapted from the Technology Acceptance Model for Mobile Services (TAMM)[89] are administered by email to assess reasons for engagement/dropout and user-experience (eg, perceived ease-of-use and adoption, perceived value, trust, acceptability and usability). Digital literacy is assessed at 1 month using the eHealth Literacy Scale (eHEALS).[90] Twelve focus groups were conducted (four in each country) at 6 months, by independent researchers, to thematically explore contextual factors (eg, influence of wider social networks), user experience and the toolkit use context in the three countries.[91] Metadata on reach (participant characteristics), dose, fidelity and engagement with the toolkit are extracted from log

files (online supplementary file 2). Methods and analyses will be detailed in separate publications.

## Data management

A NoHoW data-hub was developed by the James Hutton Institute (Edinburgh) and led by GH, which collates, monitors and stores anonymised data securely from each centre. Data from each centre are entered into trial management software (Easy Trial: www.easytrial. net). Quality checks are performed by all centres and reviewed during weekly trial management meetings. Separate publications will detail the technologies to host the toolkit, and the ICT architecture linking the toolkit, trial administration portal, trial management software and NoHoW data-hub.

## Data analysis

The 2×2 factorial design will be used in a model to investigate links between outcomes, and the extent of mediation or moderation by covariates. For continuous outcomes, linear regression models will be used. The effects of different treatment arms will be presented with 95% CI of coefficients and p values. For analyses that do not meet model assumptions, outcomes will be transformed prior to inferential comparisons. For categorical outcomes, logistic regression or categorical response models such as ordinal or multinomial regression will be used. Treatment effects will be presented as ORs, 95% CIs and p values. Missing values will be imputed using multiple imputation as part of an intention-to-treat (ITT) analysis.

To determine the mechanisms that underlie intervention efficacy, linear models are used to test whether the treatment (independent variable) causes a change in the potential mechanisms of change (mediators), which, in turn, cause a change in the primary and secondary outcomes (dependent variables). Models also investigate the effects of covariates that may moderate the effect of the interventions. Although a repeated measures/mixed model might potentially detect treatment effects not apparent when 6-month and 12-month outcomes are considered separately, our choice was driven by the primary role of the 12-month outcome. To be considered worthwhile, a WL or maintenance effect of treatment must be detectable at 12 months, whatever might happen during the time interval between then and baseline. Because of this, we chose to define the primary outcome and its analysis at 12 months alone.

For the primary outcome of weight, we intend to present results from an analysis of completers, of compliant participants and of the ITT population. Baseline and last observation carried forward are likely to be optimistic for dropouts, implying full WLM. So, for these dropouts, we will examine analyses which assume regain of all or a range of proportions of the weight lost since highest weight (in the 12 months before enrolment, for which all participants have provided evidence). Again, assuming return to baseline may be optimistic, but we do not have values at the time of greatest weight. Accordingly, we

plan multiple imputation based on the imputed weight changes described above, using also centre and gender. This is not available for other health outcomes, such as HbA1c or blood pressure.

Mediation is assessed where change in weight, as represented by a categorical endpoint (maintenance vs weight regain), is the dependent variable, intervention group is the predictor variable, and self-regulatory, emotion regulatory and other psychosocial variables are mediators. Hybrid models, combining latent and observed variables, may be specified to ensure sufficient power to conduct the analysis and to stay within the limits of sample size requirements. Model fit will be assessed using the $\chi^2$ goodness-of-fit test, Bollen's incremental fit index (0.95) and the standardised root mean residual (0.08) values. Tests of mediation will be conducted using a resampling method. To test the significance of specific indirect effects of multiple mediators (total and specific for each mediator, using both normal theory and bootstrap procedures), a tool developed by Selig and Preacher[92] will be used.

## Subgroup analysis

Interactions between condition factors and baseline participant groupings (centre, gender, age, BMI at entry, % WL at entry) will be prioritised as a priori subgroup analyses in the primary and secondary outcomes. Age, BMI and weight will be analysed as categorical variables to avoid assumptions of linearity (or the complexity of fitting polynomial terms) and for ease of presentation. These pre-planned analyses will be conducted by the inclusion of appropriate interaction terms in the regression models. Where interactions are significant, results for each subgroup will be reported. This will mitigate the problems of multiple testing due to many subgroup comparisons. We recognise that statistical power in subgroups will vary with sample size and provide a plot of detectable effect size versus sample size for reference (figure 3).

## Cost-effectiveness analyses

Cost-effectiveness analysis will be conducted to model costs and intervention effects for a lifetime horizon. An existing decision-analytical model will be used to estimate BMI category transitions, incidence of secondary diseases (eg, type 2 diabetes), quality of life and healthcare costs.[93] Quality-adjusted life-years and costs will allow calculation of incremental cost-effectiveness ratios and estimates of cost-utility. Extensive sensitivity analyses will test the modelling assumptions. Exploration of scenarios relating inter alia to the cost and effects (including sustainability) of the intervention. Secondary cost-effectiveness analysis will explore WLM outcomes (eg, cost per g maintained per year). The model will be adapted separately for each country.

## Power calculations and sample size estimation

Power calculations were based on the primary outcome (weight change). To detect a difference between

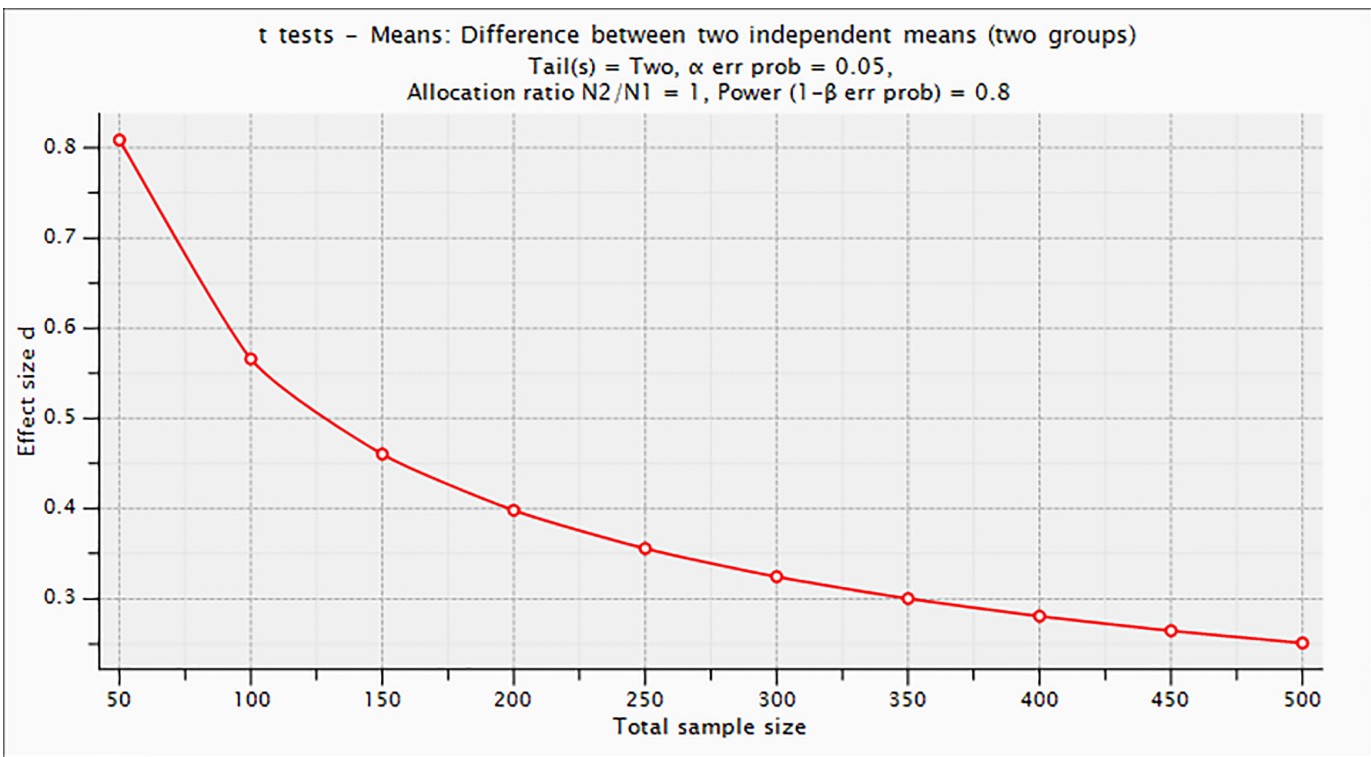

**Figure 3** Plot of detectable effect size (Cohen's d).

treatment arms of >1.5 kg body weight gives a (Cohen's) d value of 0.25.[22] For 80% power, comparing more than two groups requires a sample size of 250 per trial arm. To detect an effect size of 0.25 SD units for HbA1c, 245 participants in each trial arm gives 80% power, at 5% significance. Assuming 38% dropout,[83] a sample size of 1600 (533 per centre) is needed to achieve a sample of 1002 (334 per centre, ~250 per trial arm) participants at 12 months.

### Ethics and dissemination
The trial was registered with the ISRCTN registry (ISRCTN88405328). Ethical approval was granted by each institutional ethics committee before study commencement at each centre. The protocol complies with relevant EC legislation, international conventions and declarations relating to ethical research practices.[94] Interventions will be discontinued if they are reported detrimental. Protocol deviations, violations and serious adverse events or adverse events are recorded by trial staff and monitored by the principal investigator (RJS) and TSC.[95] Any protocol amendments that impact the study conduct are approved by each institutional ethics committee and communicated to participants. Trial results will be published in peer-reviewed scientific journals. Baseline data will be analysed after the end of recruitment (March 2018) and outcome analyses after 1-year data are collected (March 2019). Authorship will be determined by the guidelines set out by the International Committee of Medical Journal Editors.

### POTENTIAL OUTCOMES AND IMPACT
This is the first project to develop and formally evaluate (via randomised controlled trial) a new ICT toolkit combining continuous tracking of physical activity and body weight with theoretically informed, evidence-based digital interventions targeting self-regulation and motivation, and CB-emotion regulation in a 2×2 design, to target long-term weight management.

The NoHoW toolkit comprises evidence-based behaviour change techniques and CB-emotion regulation strategies, which are hypothesised mechanisms of action that may impact EB behaviours and longer term weight outcomes. Analyses of primary outcomes will enable us to test the hypotheses that self-regulation of EB behaviours improves longer term weight outcomes and that CB-emotion regulation strategies help prevent weight relapse. Secondary outcomes of the trial moderator effects will enable us to determine if different psychological and behavioural profiles of participants predict response to self-regulation or emotion regulation interventions. Mediation analyses will enable us to determine the mechanisms of action by which self-regulation and emotion regulation-based behaviour change approaches affect EB behaviours and longer term weight and health. Secondary analyses will inform more personalised digital interventions that better match intervention content and delivery to individual user needs, potentially helping to improve user engagement, health and well-being of participants.

## Limitations

Participants are blinded to study allocation, but they can access similar interventions from other real-world sources. The 2×2 design involved an active control. A 2×2 plus one design including a no-intervention control would have enabled evaluation of self-tracking using Fitbit devices versus no intervention.

The active control arm includes self-monitoring of physical activity and weight but the inclusion of the Fitbit devices across all arms potentially contaminates or diminishes the effect of self-regulatory components.

There is a potential conflict inherent in the design of trial that seeks to allow a certain degree of personalisation of intervention content use and maintain the integrity of trial arms. For this reason, the degree of personalisation was constrained to choice in the use of available self-regulatory or contextual emotion regulatory content within respective trial arms.

Social support was deliberately not part of this intervention but it is known that the degree of social support can be important for weight management.

**Author affiliations**
[1]School of Psychology, Faculty of Medicine and Health, University of Leeds, Leeds, UK
[2]Center for Research in Neuropsychology and Cognitive Behavioral Intervention, University of Coimbra, Coimbra, Portugal
[3]Faculty of Human Kinetics, University of Lisbon, Lisbon, Portugal
[4]Institute of Health & Society, Newcastle University, Newcastle upon Tyne, UK
[5]VTT, VTT Technical Research Centre of Finland, Ltd, Espoo, Finland
[6]Research Unit for Dietary Studies, Bispebjerg and Frederiksberg Hospital, The Parker Institut, Frederiksberg, The Capital Region, Denmark
[7]Department of Public Health, University of Copenhagen, Copenhagen, Denmark
[8]BioSS, Biomathematics and Statistics Scotland, Aberdeen, UK
[9]Trinity Centre for Practice and Healthcare Innovation & ADAPT Centre, University of Dublin Trinity College, Dublin, UK
[10]College of Life and Natural Sciences, University of Derby, Derby, UK

**Contributors** RJS, BLH, PJT, FFS, ALP and GH conceived the study. RJS is the principle investigator; BLH is the grant coordinator; and SES and CD are the trial managers. SCL, ALP and SES are site coordinators. GH is the trial statistician and leads data management. MH and EM are responsible for the technological development and management of the NoHoW digital toolkit. ALP, MMM, MM and CD developed the content of the arms of the digital toolkit. JE, IS, MLM, CS, BP and LR conducted the trial activities at each respective trial sites. EHE and KS were involved in the development of some of the trial outcome measures. SES, CD and RJS drafted the manuscript and all authors revised and approved the final version. All authors made substantial contributions to the conceptualisation of the study design and conduct of the protocol.

**Funding** This project has received funding from the European Union's Horizon 2020 research and innovation programme under grant agreement No. 643309. The material presented and views expressed here are the responsibility of the author(s) only. The EU Commission takes no responsibility for any use made of the information set out. MMM is funded by the Marie-Sklodowska-Curie (EDGE) Fellowship programme (grant agreement No. 713567).

**Competing interests** RJS consults for Slimming World through the University of Leeds. MMM and GH also conducted consultancy work with Slimming World. Slimming World were a partner of the project to aid recruitment within the UK but will not be involved in the analysis and interpretation of the trial.

**Patient consent for publication** Not required.

**Provenance and peer review** Not commissioned; externally peer reviewed.

**Data availability statement** All data relevant to the study are included in the article or uploaded as supplementary information.

**ORCID iD**
Sarah Ellen Scott http://orcid.org/0000-0002-8168-7064

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
