## [Reviewer comments · BMJ Open]

ARTICLE DETAILS

TITLE (PROVISIONAL)	The NoHoW protocol: A multicenter 2x2 factorial randomized controlled trial investigating an evidence-based digital toolkit for weight loss maintenance in European adults.
AUTHORS	Scott, Sarah; Duarte, Cristiana; Encantado, Jorge; Evans, Elizabeth; Harjumaa, Marja; Heitmann, Berit; Horgan, Graham; Larsen, Sofus; Marques, Marta; Mattila, Elina; Matos, Marcela; Mikkelsen, Marie-Louise; Palmeira, António; Pearson, Beth; Ramsey, Lauren; Sainsbury, Kirby; Santos, Inês; Sniehotta, Falko; Stalker, Carol; Teixeira, PJ; Stubbs, James

VERSION 1 – REVIEW

REVIEWER	Anne Moorhead Ulster University, Northern Ireland, UK
REVIEW RETURNED	29-Mar-2019

GENERAL COMMENTS	Overall - This is an excellent and comprehensive paper outlining the NoHoW protocol: A multicenter 2x2 factorial randomized controlled trial investigating an evidence-based digital toolkit for weight loss maintenance in European adults, which was very well written. This paper is appropriate for publication in BMJ. This project has received funding from the European Union's Horizon 2020 research and innovation programme under grant agreement No 643309. This trial has been registered. Abstract - very clear and excellent. Background - very good and clear with the key literature included, and underpinned by theory. Methods - The methods were appropriate and very clear. The study is an 18 month, 3-center, 2 x 2 factorial single-blind, 12 randomized controlled trial, which recruited 1627 participants achieving $\geq 5\%$ WL between March 2017 to March 2018. The NoHoW trial tests the efficacy of a 7 theoretically informed, evidence-based digital toolkit using a mobile enabled website, activity 8 trackers and Wi-Fi scales for WLM aiming to target: (i) self-regulation and motivation, and (ii) emotion regulation in adults who achieved clinically significant ($\geq 5\%$) WL in the previous 12 months (initial BMI ≥ 25 kg/m²). It was good to see cost effectiveness and public involvement included but could have been more detailed.
---

	Minor changes required: Although the primary outcome was clearly stated in the abstract - "The primary outcome is change in weight (kg) at 12 months from baseline", in the methods section - primary outcome, just "weight" was stated. In the methods section - the primary outcome needs to be clearly stated - "change in weight (kg) at 12 months from baseline" More limitations of the trial needs to be clearly stated and in more detail. Public involvement - little more detail could have been included. Potential outcomes and clinical importance of this trial could have been clearly stated at the end end of the paper
--	--

REVIEWER	Patricia A. Hageman, PhD, FAPTA University of Nebraska Medical Center 984420 Nebraska Medical Center Omaha, NE 68198-4420
REVIEW RETURNED	26-Apr-2019

GENERAL COMMENTS	The NoHoW protocol as written addresses a major public health challenge of sustaining weight loss over time. The protocol objectives are to evaluate a digital toolkit for attaining weight loss maintenance under 4 conditions 1) self-regulation and motivation, 2) emotion regulation, 3) these factors in combination, and 4) control, over an 18 month period. Participants are adults with documented verification of attaining a clinical significant weight loss of > 5% during the previous 12 months prior to enrollment. The NoHoW trial weight loss maintenance intervention is 18 months, from baseline to 6 months includes the active weight loss maintenance intervention, after which there is a 12 month period of follow-up (months 6-18 months), noting that the primary outcome is change at weight is set at 12 months. The strengths of the study include theoretical background for justifying the intervention strategies, the trial design, and inclusion/exclusion criteria with a large sample of individuals (to account for up to 38% attrition over the 18 months). Public involvement was included, which adds merits to informing the protocol development. The trial protocol as written is congruent with the published protocol on the ISRCTN registry and includes reporting of the guidelines as established by SPIRIT. The dates of participant recruitment are noted March 2017-March 2018, noting the date is past for collecting the primary outcome of change at weight at 12 months, yet the final data will not be collected until September 2019. The protocol includes clear description of data collection methods and participant retention efforts. The statistical methods appear appropriate for the intended objectives. Data monitoring was discussed. In summary, this well-written protocol addresses a topic of importance and employs a design that addresses the stated objectives.
--

REVIEWER	Hongyan Xu Augusta University
REVIEW RETURNED	23-May-2019

GENERAL COMMENTS	The design of the clinical trial is solid in the randomization scheme. The statistical analysis plan is reasonable.  1. The author should present the power analysis to justify the sample size in the clinical trial for the proposed subgroup analysis. Also the current power analysis is based on HbA1c. It's better to perform the analysis with the primary endpoint, weight loss. 2. The participant knows the intervention so it may not be called blind.
--

REVIEWER	Joseph Rigdon Senior Biostatistician Quantitative Sciences Unit Stanford University School of Medicine
REVIEW RETURNED	04-Jul-2019

GENERAL COMMENTS	This is a very well-planned study. I'm excited to see what the results are! A few comments regarding the statistics:  1. I like the simplicity of the analysis plan - linear regression on change scores - but am curious why you chose this over a repeated measures (mixed effects) model? I think both approaches are valid and am always curious how the choice is made in practice: http://journals.plos.org/plosone/article?id=10.1371/journal.pone.0006624 http://www.sciencedirect.com/science/article/pii/S0899900713000853 2. You anticipate missing data as this is a weight loss study. The mixed model would directly deal with missing data (by using the incomplete data) under a missing at random assumption without needing multiple imputation. Traditional multiple imputation also will operate under a missing at random assumption (unless you explicitly address not missing at random data using, e.g., a pattern mixture model). As I said in point 1, I think both approaches (mixed effects and multiple imputation) are valid here. I'm curious - what covariates do you plan to use in the multiple imputation? Perhaps provide a bit more on how the multiple imputation model will look. 3. For the subgroup analysis, I'm curious whether or not you will split continuous variables (e.g., BMI) into categories, or keep them as continuous. Perhaps a bit more detail here will help the reader so that everything is planned up front. 4. As I read the power calculation, your planned effect size is 0.25, which, given the anticipated -1.5kg weight change, would assume a standard deviation of weight change of 6.0kg. I see where -1.5kg comes from in ref 22, but does ref 22 also contain any info about standard deviation?
--

VERSION 1 – AUTHOR RESPONSE

Reviewer: 1

Reviewer Name: Anne Moorhead

Although the primary outcome was clearly stated in the abstract - "The primary outcome is change in weight (kg) at 12 months from baseline", in the methods section - primary outcome, just "weight" was stated. In the methods section - the primary outcome needs to be clearly stated "change in weight (kg) at 12 months from baseline".

We revised the methods section and now refer to the primary outcome as Change in weight (kg) at 12 months from baseline.

More limitations of the trial needs to be clearly stated and in more detail.

We have now expanded this section to read as follows:

“Participants are blinded to study allocation, but they can access similar interventions from other real-world sources.

Participants are blinded to study allocation, but they can access similar interventions from other real-world sources. The 2 x 2 design involved an active control. A 2 x 2 plus 1 design including a no-intervention control would have enabled evaluation of self-tracking using Fitbit® devices, versus no intervention.

The active control arm includes self-monitoring of physical activity and weight but the inclusion of the Fitbit® devices across all arms potentially contaminates or diminishes the effect of self-regulatory components.

There is a potential conflict inherent in the design of trial that seeks to allow a certain degree of personalisation of intervention content use and maintain the integrity of trial arms. For this reason the degree of personalization was constrained to choice in the use of available self-regulatory or contextual emotion regulatory content within respective trial arms.

Social support was deliberately not part of this intervention but it is known that degree of social support can be important for weight management”.

Limitations were also added at the end of the manuscript.

Public involvement - little more detail could have been included

We have now expanded this section to read as follows

“Members of the public that matched the study’s inclusion criteria at the three trial centres were consulted via a survey to inform the recruitment strategies, trial conduct, questionnaires, assessment of burden and reviewing or testing the intervention content at all trial centers. During the toolkit development stage, the public were involved in two ways. First, a panel of 10 people, matching the trial criteria, tested the toolkit version 1 written in English for one month in the UK. A mixed methods approach was used for the analysis of the data assessing technical maturity and experiences, through questionnaires (quantitative data), interviews (qualitative data), TK usage log data, and data from wireless scales and activity trackers (Fitbit). These analyses will be published in a separate paper. The results were used to inform Toolkit version 2. Toolkit version 2 was then tested for 18 months using a panel of 30 people (10 in each centre) starting three months before the launch of the trial. Trial staff regularly collaborated with the panel to identify technological glitches that could be corrected early on so not to impact on the main trial participant.”

Potential outcomes and clinical importance of this trial could have been clearly stated at the end of the paper.

We have now added a section on potential outcomes and clinical importance to the end of the manuscript, which reads “This is the first project to develop and formally evaluate (via RCT) a new ICT toolkit combining continuous tracking of physical activity and body weight with theoretically informed, evidence-based digital interventions targeting self-regulation and motivation, and emotion regulation in a 2 x 2 design, to target long term weight management.

The NoHoW toolkit comprises evidence-based behavior change techniques and emotion regulation strategies which are hypothesized mechanisms of action that may impact on energy balance behaviours and longer-term weight outcomes. Analyses of primary outcomes will enable us to test the hypotheses that self-regulation of energy balance behaviours improves longer-term weight outcomes and that emotion regulation strategies help prevent weight relapse. Secondary outcomes of the trial moderator effects will enable us to determine if different psychological and behavioral profiles of participants predict response to self regulation or emotion-regulation interventions. Mediation analyses will enable us to determine the mechanisms of action by which self-regulation and emotion regulation-based behavior change approaches affect energy balance behaviours and longer term weight and health. Secondary analyses will inform more personalized digital interventions that better match intervention content and delivery to individual user needs, potentially helping to improve user engagement, health and wellbeing of participants.

Reviewer: 3

Reviewer Name: Hongyan Xu

The design of the clinical trial is solid in the randomization scheme. The statistical analysis plan is reasonable.

1. The author should present the power analysis to justify the sample size in the clinical trial for the proposed subgroup analysis. Also the current power analysis is based on HbA1c. It's better to perform the analysis with the primary endpoint, weight loss.

The power calculation was based on weight change, and we have reworded that paragraph to make that clear. For most subgroups (gender, age group, weight loss group) we could not control the balance, but we could for trial centre, each with about 1/3 of the total sample size, implying a detectable effect size of 0.43 (Cohen's D) independently at each centre.

2. The participant knows the intervention so it may not be called blind.

Participants were aware that this is a weight loss maintenance trial but eligible participants were blind to intervention arm allocation.

Reviewer: 4

Reviewer Name: Joseph Rigdon

1. I like the simplicity of the analysis plan - linear regression on change scores - but am curious why you chose this over a repeated measures (mixed effects) model? I think both approaches are valid and am always curious how the choice is made in practice:

<http://journals.plos.org/plosone/article?id=10.1371/journal.pone.0006624>

<http://www.sciencedirect.com/science/article/pii/S0899900713000853>

We have now added the following section to the data analysis section:

“Although a repeated measures / mixed model might potentially detect treatment effects not apparent when 6 and 12 month outcomes are considered separately, our choice was driven by the primary role of the 12 month outcome. To be considered worthwhile, a weight loss or maintenance effect of treatment must be detectable at 12 months, whatever might happen during the time interval between then and baseline. Because of this, we chose to define the primary outcome and its analysis at 12 months alone”.

2. You anticipate missing data as this is a weight loss study. The mixed model would directly deal with missing data (by using the incomplete data) under a missing at random assumption without needing multiple imputation. Traditional multiple imputation also will operate under a missing at random assumption (unless you explicitly address not missing at random data using, e.g., a pattern mixture model). As I said in point 1, I think both approaches (mixed effects and multiple imputation) are valid here. I'm curious - what covariates do you plan to use in the multiple imputation? Perhaps provide a bit more on how the multiple imputation model will look.

We do not expect missing outcome data to be missing at random. For the primary outcome of weight, we intend to present results from an analyses of completers, of compliant participants and of the ITT population, which requires some data values for everyone. Baseline and last observation carried forward are likely to be optimistic for dropouts, implying full weight loss maintenance. So, for these dropouts, we will examine analyses which assume regain of all or a range of proportions of the weight lost since highest weight (in the 12 months before enrolment, for which all participants have provided evidence). This is not available for other health outcomes, such as HbA1c or blood pressure. Again, assuming return to baseline may be optimistic, but we do not have values at the time of greatest weight. Accordingly, we plan multiple imputation based on the imputed weight changes described above, using also centre and gender.

We have now expanded this section to read: "For the primary outcome of weight, we intend to present results from an analyses of completers, of compliant participants and of the ITT population. Baseline and last observation carried forward are likely to be optimistic for dropouts, implying full weight loss maintenance. So, for these dropouts, we will examine analyses which assume regain of all or a range of proportions of the weight lost since highest weight (in the 12 months before enrolment, for which all participants have provided evidence). Again, assuming return to baseline may be optimistic, but we do not have values at the time of greatest weight. Accordingly, we plan multiple imputation based on the imputed weight changes described above, using also centre and gender. This is not available for other health outcomes, such as HbA1c or blood pressure".

3. For the subgroup analysis, I'm curious whether or not you will split continuous variables (e.g., BMI) into categories, or keep them as continuous. Perhaps a bit more detail here will help the reader so that everything is planned up front.

Yes, we plan to split BMI, age and weight loss into categories. This is to avoid assumptions of linearity (or the complexity of fitting polynomial terms) and for ease of presentation.

We now state that in the Subgroup analysis section: "Age, BMI and weight will be analyzed as categorical variables, to avoid assumptions of linearity (or the complexity of fitting polynomial terms) and for ease of presentation."

4. As I read the power calculation, your planned effect size is 0.25, which, given the anticipated -1.5kg weight change, would assume a standard deviation of weight change of 6.0kg. I see where -1.5kg comes from in ref 22, but does ref 22 also contain any info about standard deviation?

This SD was based on a variety of studies which consortium members had been involved in, such as <https://trialsjournal.biomedcentral.com/articles/10.1186/s13063-015-0931-7> and <http://www.diogenes-eu.org/>

We have made no amendment to the text in relation to this point.

VERSION 2 – REVIEW

REVIEWER	Hongyan Xu Augusta University, United States
REVIEW RETURNED	21-Aug-2019

GENERAL COMMENTS	This is a revision. The authors have addressed my previous concerns.
--

REVIEWER	Joseph Rigdon Quantitative Sciences Unit Stanford University School of Medicine Palo Alto, CA, USA
REVIEW RETURNED	13-Aug-2019

GENERAL COMMENTS	Very thorough presentation of study design, methods, and planned statistical analyses. I look forward to reading the results!
---